# Research on Quantitative Analysis of Multiple Factors Affecting COVID-19 Spread

**DOI:** 10.3390/ijerph19063187

**Published:** 2022-03-08

**Authors:** Yu Fu, Shaofu Lin, Zhenkai Xu

**Affiliations:** 1Faculty of Information Technology, Beijing University of Technology, Beijing 100124, China; 13138109993@163.com (Y.F.); vaxwan1314@163.com (Z.X.); 2Beijing Institute of Smart City, Beijing University of Technology, Beijing 100124, China

**Keywords:** quantitative analysis, COVID-19, Gauss-Newton iteration, neural network

## Abstract

The Corona Virus Disease 2019 (COVID-19) is spreading all over the world. Quantitative analysis of the effects of various factors on the spread of the epidemic will help people better understand the transmission characteristics of SARS-CoV-2, thus providing a theoretical basis for governments to develop epidemic prevention and control strategies. This article uses public data sets from The Center for Systems Science and Engineering at Johns Hopkins University (JHU CSSE), Air Quality Open Data Platform, China Meteorological Data Network, and WorldPop website to construct experimental data. The epidemic situation is predicted by Dual-link BiGRU Network, and the relationship between epidemic spread and various feature factors is quantitatively analyzed by the Gauss-Newton iteration Method. The study found that population density has the greatest positive correlation to the spread of the epidemic among the selected feature factors, followed by the number of landing flights. The number of newly diagnosed daily will increase by 1.08% for every 1% of the population density, the number of newly diagnosed daily will increase by 0.98% for every 1% of the number of landing flights. The results of this study show that the control of social distance and population movement has a high priority in epidemic prevention and control strategies, and it can play a very important role in controlling the spread of the epidemic.

## 1. Introduction

Since December 2019, The Corona Virus Disease 2019 (COVID-19) caused by the SARS-CoV-2, has spread rapidly around the world. On 11 March 2020, the WHO announced that COVID-19 has become a major issue in the world [1,2,3,4]. The spread of COVID-19 has had a serious impact on the medical and economic aspects of countries around the world [5]. Due to the complexity of the spread of COVID-19, existing models cannot accurately estimate the direction of the spread of the epidemic [6]. Therefore, we need to build a quantitative analysis model to deeply explore the spread and influencing factors of COVID-19 on a global scale. In the current research, the data-driven deep learning model has an outstanding performance in the task of modeling time series [7].

The symptoms of COVID-19 are fever, cough, shortness of breath, loss of consciousness and fatigue. Other symptoms include dyspnea and chest pain [8]. In order to prevent the spread of the epidemic, countries have adopted many measures, such as reducing gathering activities, controlling the movement of people, advocating the use of masks, and regular disinfection in public areas [9]. As of 31 December 2021, there have been more than 287 million confirmed cases of COVID-19 worldwide, and at least 5 million people have lost their lives [10]. In order to further grasp the factors affecting the spread of SARS-CoV-2, better support the decision-making of epidemic prevention and control, timely made targeted countermeasures, and control the further spread of the epidemic, it is very urgent to quantitatively analyze the relationship between various factors and the spread of SARS-CoV-2.

The remainder of this paper is arranged as follows. Section 2 comprehensively introduces the current research on COVID-19 and the transmission characteristics of the SARS-CoV-2. Section 3 introduces the data sources and presents our research methodology. Section 4 describes the experimental results and provides an analytical discussion, and Section 5 summarizes the conclusions of this study and proposes further research directions.

## 2. Related Research Work

### 2.1. Research on COVID-19 Epidemic

Since COVID-19 outbreak in December 2019, research on COVID-19 has attracted the attention of data scientists from all over the world. Duccio et al. [11] predicted that the maximum number of infections in Italy was about 26,000 and the death toll was about 18,000 through analysis of the spread of the epidemic in China and France. Ricardo et al. [12] proposed a regression of compressed space Gaussian processes based on chaotic dynamics system to predict the number of people infected with COVID-19 in the United States, and concluded that the number of infected people in the United States would reach more than one million on 14 June 2020. Rohit et al. [13] proposed Genetic Evolutionary Programming (GEP) to analyze and predict the amount of COVID-19 cases in India. They proposed a GEP model based on the use of a simple function, which was highly effective for the time series prediction of COVID-19 cases in India. Putra et al. [14] used Particle Swarm Optimization (PSO) to estimate the parameters in the Susceptible Infectives Recovered Model (SIR), and concluded that the parameter results of the PSO algorithm were more accurate and had lower errors than the traditional method. Mbuvha et al. [15] estimated the parameters of the SIR with data from Lombardy, Italy and Hubei, China, and used the SIR model to predict the number of COVID-19 cases in South Africa, and concluded that COVID-19 was still in the early stage in South Africa.

So far, some scholars have done excellent research, but if it is necessary to further study the transmission characteristics of the SARS-CoV-2, it is impossible to predict the number of patients only. It is necessary to collect data related to the spread of SARS-CoV-2, and to analyze the characteristics of SARS-CoV-2 to understand what factors are related to the spread of SARS-CoV-2 and the quantitative relationship between them, so as to support the more precise adoption of effective prevention, control and disposal measures.

### 2.2. Research on the Transmission Characteristics of the SARS-CoV-2 Virus

When COVID-19 became a global hot topic, people put forward many speculations that could affect the transmission characteristics of the SARS-CoV-2, such as temperature [16,17,18], humidity [19,20], population density [21,22], age [23,24], and so on. In this regard, scholars have also conducted a lot of research, which has a non-negligible inspiration for us to reveal the transmission characteristics of the SARS-CoV-2. Lin et al. [25] studied the relationship between climate and the spread of COVID-19 on a global scale, and concluded that the spread of COVID-19 was highly correlated with temperature and relative humidity. Roengrudee et al. [26] studied the relationship between smoking and the spread of COVID-19, and concluded that there was a significant correlation between the number of smokers and the spread of COVID-19. Kass et al. [27] analyzed the relationship between Body Mass Index (BMI) and age in the number of confirmed COVID-19 patients through a multiple linear regression model, and concluded that obesity may increase the infection rate of COVID-19. WU et al. [28] found that in the United States, areas with higher historical PM2.5 were positively correlated with higher COVID-19 mortality. Hamit et al. [29] found that population density was the main factor affecting the spread of the epidemic through research on the spread of the epidemic in Turkish cities.

The above-mentioned studies generally have the following problems: (1) The area covered by the data set is limited to local areas, and the propagation characteristics of SARS-CoV-2 cannot be analyzed from a global scale. (2) The conclusion is only a qualitative analysis, and it has not been able to quantify the effects of various factors on the impact of the spread of the SARS-CoV-2. In response to the above problems, this paper constructs a quantitative analysis model between COVID-19 and multiple factors. Firstly, we collect the required data on a global scale, and then build a Dual-link BiGRU prediction network to predict the number of new cases in each country every day, and quantitatively analyze the impact of different factors on the number of new cases per day of COVID-19. Compared with the above research, the model proposed in this paper is more helpful to analyze the development trend of the epidemic on a global scale, helps to grasp the characteristics of the SARS-CoV-2, and provides more clear theoretical support for the subsequent formulation of anti-epidemic policies by governments of various countries.

## 3. Data Sources

The data set in this paper is mainly divided into four parts including epidemic data, climate data, population and flight data, and air quality data.

The source of the epidemic data is COVID-19 data set published by the Center for Systems Science and Engineering (CSSE) at Johns Hopkins University. The data set was collected from all over the world from 22 January 2020, in the early stage of the epidemic. The experimental data in this article include the collected epidemic data from 22 January 2020 to 31 December 2021. The feature data elements include the cumulative number of confirmed cases, the cumulative number of cured people, the cumulative number of deaths, and the number of new cases per day.The climate data comes from the daily recorded data of weather stations around the world collected by the China Meteorological Data Network (http://data.cma.cn/). This experiment selects the climate data of various regions from 22 January 2020 to 31 December 2021. The feature data elements include daily maximum temperature, daily minimum temperature, wind speed, precipitation, dew point temperature, atmospheric pressure, wind gust, altitude, absolute humidity and relative humidity.The population and flight data come from the Population Division of the Department of Economic and Social Affairs of the United Nations Secretariat. (https://population.un.org/wpp/). This experiment selects population and flight data in various regions from 22 January 2020 to 31 December 2021. The feature data elements include total population, population density, the total number of flights, number of domestic flights, and international flights.The air quality data come from the open-source air quality website WAQI (https://aqicn.org/data-platform/covid19/). This experiment selects air quality data in various regions from 22 January 2020 to 31 December 2021. The feature data elements include NO${}_{2}$, PM${}_{10}$, PM${}_{2.5}$, PM${}_{1}$, SO${}_{2}$, O${}_{3}$, CO content in the air, Air Quality Index(AQI), Suspended particle concentration(from NEPH), UV Index(UVI), Pollution(POL) and Wavelength Dominant(WD).

We collected 31-dimensional features of 81 countries to form a data set. Because we can get the data we need in these countries, we selected these 81 countries. In order to ensure that there was a sufficient amount of data to train the model, we selected the 9:1 segmentation ratio to divide the training set and test set, that is, the data from 22 January 2020 to 31 October 2021 was set as the training set and that from 1 November 2021 to 31 December 2021 as the test set.

## 4. Research Methods

The quantitative relationship model between COVID-19 spread and various characteristic factors proposed in this paper includes three steps: multi-source heterogeneous data preprocessing, constructing Dual-link BiGRU Network to prediction COVID-19 spread, and building a quantitative analysis model of multiple feature data relationships.

### 4.1. Multi-Source Heterogeneous Data Preprocessing

Because the data comes from a variety of public data sets, there are some problems among data sets, such as inaccurate data, missing data, inconsistent data format and etc. In the data preprocessing stage, this paper builds a dataset with the original data as the core. For inaccurate data, when the values of the same feature data in datasets from different sources are the same, we consider the data to be reasonable; otherwise, most of the data in datasets from different sources are selected as the final data. For missing data, the Cubic Spline Interpolation method is used to supplement the data. For inconsistent data format, feature level fusion method is adopted to extract the features of each source data set first, while the extracted feature information comes from the high-order representation of the original information, and then to aggregate and synthesize the multi-source data according to the feature information. The data with inconsistent scales are normalized by the linear normalization method to unify the data scale. This is also a commonly used data preprocessing method in the field of COVID-19 prediction. The information contained in the fused data is shown in Table 1.

### 4.2. Dual-Link BiGRU Network to Predict the Spread of COVID-19

In this paper, we construct Dual-link BiGRU Network to predict the spread of COVID-19. The task of Dual-link BiGRU is to regress and predict the number of new cases per day with input data. Dual-link BiGRU conducts parameter training through the relationship between daily different factors in the training set and the number of new cases. It inputs the values of the daily factors in the test set, and outputs the regression estimation of the number of new cases on that day. The network structure diagram of Dual-link BiGRU is shown in Figure 1.

Dual-link BiGRU consists of a dual-link feature network and a fully connected network. In the feature network, Link 1 is composed of one-dimensional convolutional network, BiGRU network, and one-dimensional inverse convolutional network. Link 2 is composed of one-dimensional convolutional network, fully connected network, and one-dimensional inverse convolutional network. Link 1 is mainly responsible for learning the timing information in the data of multiple factors. The one-dimensional convolutional network in Link 2 provides a larger receptive field for the network with a larger size of convolution kernel to learn different feature information from Link 1. In this experiment, in order to obtain a larger receptive field and better features, we select the kernel size of 16. After the dual-link feature network is a fully connected network. The fully connected network’s main function is to change the output dimension of the entire Dual-link BiGRU network to the desired output dimension.

According to the prediction performance of the test set, the parameter settings of the prediction network are shown in Table 2. The optimizer used for model training is Adam, the loss function is Mean Squared Error Loss Function (MSELoss), and the number of iterations is set to 500. In this paper, we selects BiLSTM [30], BiGRU [31], and CNN [32] for comparison at the same dataset which comes from Table 1. BiLSTM, BiGRU, and CNN are connected by their respective models and fully connected layers. The hidden layer size and number of layers of BiLSTM and BiGRU are consistent with Dual-link BiGRU, and the parameter setting of CNN is consistent with 1-D Conv1 in Dual-link BiGRU.

### 4.3. The Quantitative Analysis Model of Multi Characteristic Data Relationships

In this paper, we sets the tolerance of the prediction error rate β∈ [0, 1]. The model with a prediction error rate lower than β is called an effective model, otherwise it is called an invalid model. It is assumed that only effective models can participate in quantitative analysis. Therefore, the larger of β means the more effective models, and the quantitative analysis results have better generalization ability, but it also means that the results have larger errors; the smaller of β means the less effective the models and the poorer generalization ability of the quantitative analysis results, while the results have smaller errors within a limited range. This paper needs to have a small result errors on the basis of ensuring a certain generalization ability, so β = 0.2 is set in the experiment of this paper.

In this paper, the Gauss-Newton iterative method is used for quantitative analysis. The Gauss-Newton iterative method uses Taylor series expansion to approximately replace the nonlinear regression model. Through multiple iterations, the regression coefficient is modified many times, so that the regression coefficient continuously approaches the best regression coefficient of the nonlinear regression model, and finally the Residual Sum of Square of the original model is minimized.

According to the selected observation variable data, a multiple nonlinear regression model as in Equation (Equation 1) can be constructed.
(1)$$\begin{array}{c} y=f(X,\beta )+\epsilon \end{array}$$
where *y* is the dependent variable, which represents the number of newly diagnosed people every day in this experiment; *X* is the set of independent variables, which represents the data of each characteristic factor in this experiment; β is an unknown parameter; ϵ is an error term, and it is an unobservable random variable with a mean of zero and a variance of $\sigma^{2}>0$. The above model can be used to predict the number of the newly diagnosed daily and determine the nonlinear quantitative relationship between each independent variable and the dependent variable. The Gauss-Newton iteration method estimates the to-be-regressed parameter β of the nonlinear regression model through continuous iteration.

The realization process of the quantitative analysis model includes the following steps:Construct multiple regression models and train through data;The prediction ability of the model is evaluated by modifying the determination coefficient;The quantitative relationship between multiple factors and the number of new cases per day was determined by a multiple regression model;Given different initial values for different factors $x_{0}$;For the kth iteration, calculate the Jacobian matrix *J*, Hessian matrix *H*, *B*, and calculate the increment $△x_{k}$;If $△x_{k}$ is small enough, stop the iteration, otherwise, update $x_{\left(k+1\right)}$ = $x_{k}$ + $△x_{k}$;Repeat steps (5) (6) until the maximum number of iterations is reached, or the termination condition of (6) is met;Complete the estimation of the unknown parameter β, and determine the quantitative relationship between different elements and the number of new cases per day;Complete for β to determine the quantitative relationship between different elements and the number of new cases per day.

## 5. Experimental Results and Discussion

### 5.1. Dual-Link BiGRU

In this paper, the evaluation index is selected as the error rate ρ, and the error rate calculation formula is shown in Equation (Equation 2):(2)$$\rho =\left(1/m\sum_{i=0}^{m}\left({\hat{y}}_{i}-y_{i}\right)\right)/\left(1/m\sum_{i=0}^{m}y_{i}\right)$$
where ${\hat{y}}_{i}$ represents the model output, $y_{i}$ represents the label of the number of new cases per day, and m represents the total number of samples in the test set. This indicator can measure the gap between the model output and the label of the entire test set sample.

In this paper, we selects BiLSTM [30], BiGRU [31], and CNN [32] for comparison at the same dataset which comes from Table 1. BiLSTM, BiGRU, and CNN are connected by their respective models and fully connected layers. The hidden layer size and number of layers of BiLSTM and BiGRU are consistent with Dual-link BiGRU in Table 2, and the parameter setting of CNN is consistent with 1-D Conv1 in Dual-link BiGRU in Table 2. Sets the prediction error tolerance β = 0.2, and uses the model error rate as the evaluation index. In the data of 81 countries, the model with an error rate lower than β is regarded as an effective model, and the difference in the number of effective models among different models is compared in the test dataset. The comparison experiment results are shown in Table 3.

Table 3 shows that (1) Dual-link BiGRU has a larger effective model ratio in the prediction network; (2) Compared with BiGRU, BiLSTM, and CNN, Dual-link BiGRU performs better in low error rate. Therefore, it is believed that the Dual-link BiGRU has better performance and generalization ability in predicting the daily number of new epidemics in various countries. Therefore, this paper selects the Dual-link BiGRU as the prediction network. Figure 2 shows the difference between the daily number of new cases predicted of the Dual-link BiGRU and the label value. Because showing the forecast results for all countries would make the paper extraordinarily long, in this paper, we select 6 countries with better results for display, including Canada, China, India, Indonesia, Russia, and United Kingdom.

It can be seen from Figure 2 that in the selected 6 countries, the red solid line is the label of the number of new cases per day, and the green dashed line is the predicted value by the Dual-link BiGRU network. The two curves have a high degree of overlap. Therefore, the prediction network constructed in the experiment has a good fit with the real data. The trained prediction network can better predict the daily new cases and has a strong generalization ability. For different countries, the model can learn more appropriate parameters to predict the number of the daily new cases.

### 5.2. Quantitative Analysis Results of Multi-Characteristic Data Relationships

In this paper, we uses the method of Lin [25] and others to build a multiple regression model for the selected 44 effective national models and train them. Through the multiple regression model, the quantitative relationship between multiple factors and the number of new cases per day is determined, and the prediction ability of the model is evaluated by determining the Adjusted R Square (R). The larger R is, the stronger the prediction ability of the model is. If R is greater than 0.6, the model has strong epidemic prediction ability. Then, the initial value of the Gauss-Newton iterative method is selected through the model parameters. The quantitative relationship between multiple factors and the number of new cases per day is shown in Table 4, and the initial values are shown in Table 5.

In this paper, we uses the trained Dual-link BiGRU model of various countries to generate simulation data for quantitative analysis. The data generation method is as follows:Goal: To generate data for analyzing the quantitative relationship between $x_{1}$ and *y*, where $x_{1}$ is the maximum temperature per day and y is the number of new cases per day.To control other factors unchanged, adjust $x_{1}$, and generate the predicted value of *y*.The simulation data is used as input, and training is performed with the Gauss-Newton method to obtain the coefficient between $x_{1}$ and *y*, so as to determine the quantitative relationship between them.

According to the above method, the coefficient equations between the number of new cases per day in each country and the characteristic factors in Table 1 are obtained respectively, and the quantitative relationship between the number of new cases per day and the characteristic factors in each country is determined. Then take the average of the quantitative relationship coefficients of the same feature in all countries, and finally get the quantitative relationship between each feature that is applicable in the selected country and the number of new cases per day with strong generalization performance, as shown in Table 6.

As shown in Table 6, among the selected features, the population density per unit land area has the largest positive correlation with the number of new cases per day, followed by the number of landing flights. The population density per square kilometer increases by 1%, and the number of new cases per day in the corresponding area increases by about 1.076%. For every 1% increase in the number of landing flights, the number of new cases per day in the corresponding area increases by about 0.98%. Among the selected features, the daily maximum temperature, daily minimum temperature and dew point temperature have negative correlations to the number of new cases per day. Within the range of 0–50 °C, each increase of 1 °C can reduce the number of new cases per day by 0.021%, 0.028% and 0.015% respectively.

Based on the above analysis, the following further inferences can be drawn:Population factors and flight factors has an obvious positive correlation impact on the spread of COVID-19. From the data of the selected 44 countries, it can be seen that population factors and flight factors have a greater impact on the spread of COVID-19. Every 1% increase in population factors will increase the spread of the epidemic by 1.044%. Every 1% increase in the number of arrival flights will increase the spread of the epidemic by 0.98%. Therefore it can be seen that population factors and flight factors have a more obvious impact on the increase in the spread of the epidemic. From the perspective of formulating epidemic prevention and control policies, controlling social distancing and population movement will have a more obvious positive correlation impact on epidemic prevention and control.The increase in temperature and relative humidity has a negative correlation impact on the spread of COVID-19.Among the climatic factors, the increase of temperature and humidity has a negative correlation impact on the spread of COVID-19. In this paper, the temperature range of 0–50 °C and the relative humidity range of 1–100% are selected for the experiment. It is obtained that within this range, temperature and relative humidity has a negative correlation impact on the spread of COVID-19, but the impact is not obvious. Since the correlation between population density and the speed of the epidemic is far greater than the correlation between temperature and the speed of the epidemic, it is speculated that in areas with higher temperatures and higher population densities, such as India and other countries, the speed of the epidemic still has a relatively rapid possibility.A larger AQI has a positive correlation impact on the spread of COVID-19.AQI represents the degree of air cleanliness or pollution and its impact on health. The higher the AQI, the more serious the air pollution in the region. This experiment shows that in the range of AQI value 100–200, the epidemic transmission speed of COVID-19 will increase by 0.013% every time AQI increases by 1. Some researchers have shown that SARS-CoV-2 can spread through aerosols [33,34,35]. Therefore, a higher AQI means a higher aerosol content in the air, which is not good for air circulation. Such an environment may promote the spread of COVID-19.

## 6. Discussion

Since the discovery of COVID-19 in 2019, countries have successively formulated epidemic prevention and control policies that suit their own national conditions [36]. According to the current development status of the world epidemic, a long-term coexistence with the virus has been formed, that is, even though the vaccine has been developed, it will take a long time to completely eliminate COVID-19 [37,38]. This paper carries out quantitative analysis and research on COVID-19 transmission by various factors all over the world and comes to the conclusion that the increase of population density, population flow, and flight times has a positively correlated impact on the epidemic transmission, and the increase of temperature, relative humidity, and dew point temperature has a negative correlation impact on the epidemic transmission. It can be concluded that the positive correlation effect of population density on the epidemic spread is much greater than the negative correlation effect of climate factors on the epidemic spread.

Therefore, according to the regional characteristics and national conditions, governments should formulate epidemic prevention and control policies to control population density and population flow in the climate environment with high local temperature and relative humidity, maximize the effect of epidemic prevention and control, and curb the spread of the epidemic from the aspects of transmission route and virus characteristics.

International organizations need to establish high, medium and low-risk epidemic spread levels globally. The faster the epidemic spread, the higher the epidemic spread level, and the more stringent prevention and control policies need to be adopted. For cities where the epidemic has spread, it is necessary to keep wearing masks, maintain proper social distancing, and reduce public recreational activities. For cities with large population density and serious epidemic spread, it is recommended to strictly control population flow, tighten restrictive measures for international flights, and take “city closure” measures when necessary, and other cities need to take more stringent entry epidemic prevention measures for personnel from high-risk countries and regions. For cities with slow epidemic spread, it is suggested to control the population flow within a certain range, allow international flights under the condition of good epidemic prevention measures, strictly control the flow of people from high-risk countries and regions, and be vigilant against the epidemic spread caused by climate change.

## 7. Conclusions and Future Work

In this paper, we fuses multi-source heterogeneous data, and makes predictions for the current COVID-19 epidemic based on the fusion data set, and quantitatively analyzes the model to obtain the quantitative relationship between various factors and the spread of the epidemic. The contributions of this paper are as follows:Dual-link BiGRU network is proposed, which integrates time-series features and high-order features through dual-link construction, and can obtain more accurate prediction effects and generalization capability. Through experiments, it can be determined that the Dual-link BiGRU network has the following advantages:
Compared with the CNN, LSTM, and GRU networks, the prediction accuracy of the Dual-link BiGRU network is improved by 35.03%, 31.41%, and 27.36%, respectively;Compared with the CNN, LSTM, and GRU networks, the generalization ability of the Dual-link BiGRU network is improved by 25.00%, 27.50%, and 28.75%, respectively.According to the quantitative analysis between the SARS-CoV-2 virus and its characteristic factors on a global scale, we concluded that the SARS-CoV-2 virus transmission has the following characteristics:
The increase in population factors and flight factors has an obvious positively correlated impact on the spread of COVID-19.The increase in AQI will has a minor positively correlated impact on the spread of COVID-19.The increase in temperature and relative humidity has a negative correlation impact on the spread of COVID-19.

Accordingly, this paper makes the following recommendations for global epidemic prevention and control:Countries should take appropriate or even stricter prevention and control measures according to their national conditions, such as demographic factors, climate factors, air quality factors, and the number of flights, to minimize the risk of outbreaks.Demographic factors have a strong positive relationship with the spread of COVID-19 epidemic. Governments can control the spread of the epidemic by strictly controlling the movement of people both within and outside the country.Since the impact of population and flight factors on the spread of the epidemic is much greater than that of climate factors, governments of various countries should not expect the epidemic to disappear after the temperature rises, and should actively control population movement.

This paper has completed the multi-factor quantitative analysis model affecting the spread of COVID-19. Due to the different detection coverage of COVID-19 in various countries, the number of confirmed cases is inevitably underestimated, and this paper does not evaluate the impact of changes in policies and local prevention and control strategies on the spread of COVID-19. Therefore, more detailed exploration is needed on these issues in the next step.

## Figures and Tables

**Figure 1 ijerph-19-03187-f001:**
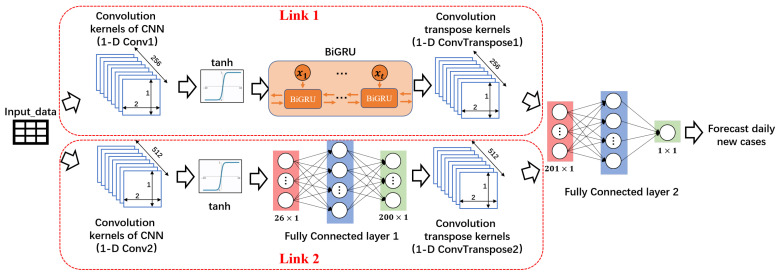
The network structure diagram of Dual-link BiGRU.

**Figure 2 ijerph-19-03187-f002:**
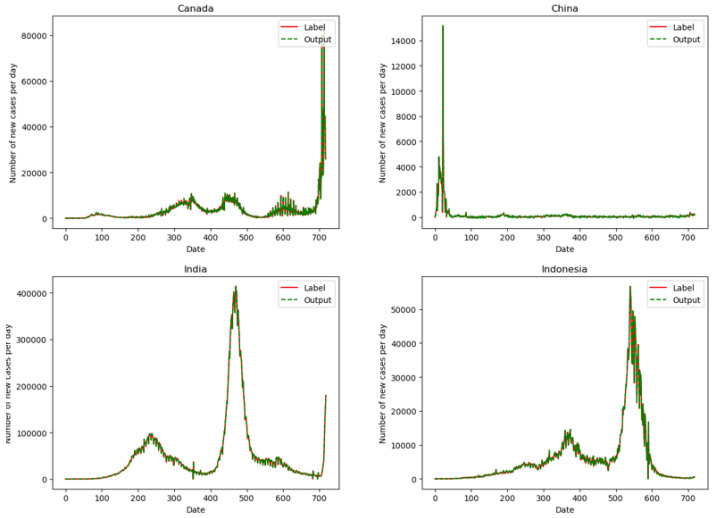
Display of Dual-link BiGRU prediction results.

**Table 1 ijerph-19-03187-t001:** Feature display of fusion data set.

Feature Category	Feature Range
**Date**	**22 January 2020–31 December 2021**
Country	Afghanistan, Algeria, Argentina, Australia, Austria, Bahrain, Bangladesh, Belgium,
	Bolivia, Brazil, Bulgaria, Canada, Chile, China, Colombia, Costa Rica,
	Croatia, Cyprus, Denmark, Ecuador, El Salvador, Estonia, Ethiopia, Finland,
	France, Georgia, Germany, Ghana, Greece, Guatemala, Guinea, Hungary,
	Iceland, India, Indonesia, Iran, Iraq, Ireland, Israel, Italy,
	Japan, Jordan, Kazakhstan, Korea, Kuwait, Kyrgyzstan, Laos, Lithuania,
	Macedonia, Malaysia, Mali, Mexico, Mongolia, Nepal, Netherlands, New Zealand,
	Norway, Pakistan, Peru, Philippines, Poland, Portugal, Romania, Russia,
	Saudi Arabia, Serbia, Singapore, South Africa, Spain, Sri Lanka, Sweden,
	Switzerland, Tajikistan, Thailand, Turkey, Uganda, Ukraine, United Arab Emirates,
	United Kingdom, United States, Uzbekistan
Epidemic	Confirmed, Recovered, Deaths, New
Climate	Tmax, Tmin, Wind_speed, Precipitation, DP_F,
	Pressure, Wind_gust, Altitude, Ab_humidity, Re_humidity
Population	Pop, Density
Air quality	NO${}_{2}$, PM${}_{10}$, PM${}_{2.5}$, PM${}_{1}$, SO${}_{2}$, O${}_{3}$, CO and AQI, NEPH, UVI, POL, WD
Flight	Flight_total, Flight_domestic, Flight_international

Tmax, Tmin, Wind_speed, Precipitation, DP_F, Pressure, Wind_gust, Altitude, Ab_humidity and Re_humidity represent daily maximum temperature, daily minimum temperature, daily average wind speed, daily rainfall, daily dew point temperature, atmospheric pressure, wind gust, altitude, absolute humidity and relative humidity. Pop, Density represent total population, population density. NO${}_{2}$, PM${}_{10}$, PM${}_{2.5}$, PM${}_{1}$, SO${}_{2}$, O${}_{3}$, CO and AQI, NEPH, UVI, POL, WD represent NO${}_{2}$, PM${}_{10}$, PM${}_{2.5}$, PM${}_{1}$, SO${}_{2}$, O${}_{3}$, CO content in the air, Air Quality Index(AQI), Suspended particle concentration(from NEPH), UV Index(UVI), Pollution(POL) and Wavelength Dominant(WD). Flight_total, Flight_domestic, and Flight_international represent the total number of flights, the number of domestic flights, and the number of international flights respectively.

**Table 2 ijerph-19-03187-t002:** Prediction network parameter settings.

Layer	Parameter	Value
1-D Conv1	Out channels	256
	Kernel size	16
	Stride size	8
1-D Conv2	Out channels	512
	Kernel size	16
	Stride size	8
BiGRU	Hidden size	100
	Number of layers	5
1-D ConvTranspose1	Out channels	256
	Kernel size	16
	Stride size	8
1-D ConvTranspose2	Out channels	512
	Kernel size	16
	Stride size	8
Full Connected layer 1	In channels	26
	Out channels	200
Full Connected layer 2	In channels	201
	Out channels	1

**Table 3 ijerph-19-03187-t003:** Comparison of model results.

Model	0–5%	5–10%	10–15%	15–20%	>20%	Effective	Invalid
Dual-link BiGRU	2	12	12	22	33	48	33
BiGRU	0	6	7	12	56	25	56
BiLSTM	0	6	8	10	57	24	57
CNN	0	7	8	12	54	27	54

**Table 4 ijerph-19-03187-t004:** Regression equation parameter.

	Confirmed	Recovered	Deaths	Tmax	Tmin
Global	0.06	0.17	−0.28	−4.52	−2.97
	Wind_speed	Precipitations	DP_F	Pressure	Wind_gust
	−16.46	84.64	−4.67	2.02	73.72
	Altitude	Ab_humidity	Re_humidity	Pop	Density
	6.71 × 10${}^{-7}$	−0.17	−0.112	5.8 × 10${}^{9}$	54,282.5
	NO${}_{2}$	PM${}_{10}$	PM${}_{2.5}$	PM${}_{1}$	SO${}_{2}$
	1.95 × 10${}^{3}$	49.42	55.59	45.29	−21.91
	O${}_{3}$	CO	AQI	NEPH	UVI
	65.56	12.61	0.14	−8.45	−1.46
	POL	WD	Flight_total	Flight_domestic	Flight_international
	23.68	1.91	189.547	379.995	187.5932
	ϵ	Adjusted R Square			
	293.18	0.79			

**Table 5 ijerph-19-03187-t005:** Example of initial value of each characteristic coefficient.

Country	Tmax	Tmin	DP_F	……	Re_Humidity	Density	Iterations
Canada	0.58	−0.91	−0.0075	……	−1.67	0.34	100
China	2.33	−11.48	−18.34	……	−12.03	0.071	100
India	−5.22	−16.35	−19.45	……	−15.50	−1.44	100
Indonesia	5.64	4.25	14.55	……	−1.15	−0.88	100
Russia	−23.36	28.45	40.13	……	−2.71	0.23	100
United Kingdom	−391.08	244.49	698.08	……	262.37	−34.67	100

**Table 6 ijerph-19-03187-t006:** Quantitative relationship between characteristic factors and daily number of new cases.

Features	Particle	Influence/%
Density	+1%/km${}^{2}$	1.0767212
Pop	+1%/km${}^{2}$	1.0441276
Flight_total	+1%	1.0102873
flight_domestic	+1%	0.9881371
flight_international	+1%	0.9455161
UVI	+1%	0.8142484
PM${}_{2.5}$	+1 μg/m${}^{3}$ in the range of 0–100 μg/m${}^{3}$	0.0126328
PM${}_{10}$	+1 μg/m${}^{3}$ in the range of 0–100 μg/m${}^{3}$	0.0124261
NO${}_{2}$	+0.3 μg/m${}^{3}$ in the range of 0–30 μg/m${}^{3}$	0.0190209
SO${}_{2}$	+0.1 μg/m${}^{3}$ in the range of 0–10 μg/m${}^{3}$	0.0208433
PM${}_{1}$	+1 μg/m${}^{3}$ in the range of 0–100 μg/m${}^{3}$	0.0145565
Wind_speed	+1 m/s in the range of 0–10 m/s	−0.0135183
Preciptation	+1%	−0.0198199
Re_humidity	+1%	−0.0159099
DP_F	+1 °C in the range of 0–50 °C	−0.0150033
Tmin	+1 °C in the range of 0–50 °C	−0.0285928
Tmax	+1 °C in the range of 0–50 °C	−0.0217991

The influence >0, indicating that the factor has a positive correlation with the increase in the number of new cases per day. The influence <0, indicating that the factor has a negative correlation with the increase in the number of new cases per day.

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
