# Peer review of "Research on Quantitative Analysis of Multiple Factors Affecting COVID-19 Spread"

_ijerph, 2022, doi:10.3390/ijerph19063187_

Round 1

Reviewer 1 Report

 In this paper, the authors use both machine learning-based and the Gauss-Newton Iteration methods to predict the covid19 case number and quantify the relationship between it and some features; based on the findings, the authors also propose some conclusions and suggestions. I think overall the results are useful given the current situation of covid, but there are some unclear things in the paper:

  1. The authors should add more descriptions about dual-link BiGRU. For example, 1) the authors used a kernel size of 16, which is much larger than the common size in machine learning. Are there any specific reasons for doing this? 2) what is the configuration of two Fully connected layers?
  2. it is confusing to plot BiGRU in the same way of plotting the Fully connected layer in Fig. 1. RNN should not be plotted like this.
  3. the authors claimed that "This paper selects BiLSTM[30], BiGRU[31], and CNN[32] for comparison, sets the prediction error tolerance b=0.2, and uses the model error rate as the evaluation index. " but did not give any further details for those individual models, which will be hard to understand if the comparisons are valid. For example, are those models comparing against the same dataset? I suggest the authors add more descriptions for those models.
  4. what dataset is being compared in table 3? I assume it is the test dataset, but it should be pointed out in the paper.
  5. Reference 31 has nothing to do with BiGRU.
  6. Figure 2 has missing x-axes in Canada and China.
  7. Why there are 365 days in Figure 2? The test dataset only contains ~60 days (from Nov 1st, 2021 to Dec 31st, 2021), while the training dataset contains way more than 365 days.

Reviewer 2 Report

I am favorable to the publication of this article, providing the following revisions:
 There are many grammar and vocabulary errors, and English should be refined.(all are highlighted)

Reviewer 3 Report

Introduction

In this paper, the authors used Gauss-Newton Iteration method to quantitatively analyze factors that affect the spread of COVID-19 virus around the world. In their work, they relied on data from all the world during the period 22 January 2020 to 31 December 2021. They have collected four types of data. The first type, epidemic data, deals with cumulative number of confirmed cases, cured people and deaths. For the second type, climate data, the authors have considered daily climate data, as daily max-min temperature, wind speed, humidity, and altitude. The third data part, population and flights, is related to population density, total number of flights, as examples. The fourth one reports air quality index and some molecules substances, for example.  

The authors’ methodology consists of three steps: preprocessing of the 31-dimensional heterogeneous data, construction of the Dual-link BiGRU network to predict the spread of COVID-19, and finally the construction of a quantitative analysis model.

The authors found that the population density, the number of landing flights are the most correlated factors to the spread of the virus. They concluded that the control of social distance and the movement of people present the highest priority measures to be followed to limit the spread of the pandemic.

Comments

  1. In the first research question of the work, the authors claim that previous literature works considered only limited regions in their studies. They also claim that their work will have a global scale. Whereas, they have used data from 81 countries, e., about half of 195 countries (the world total number of countries).
  2. The number of countries per continent is unequal: the smallest number of countries is for Africa. Additionally, Egypt, which has an important population (102 million in 2020), is missing. What is the authors’ strategy in selecting the countries?
  3. In the preprocessing phase:

-        How do the authors decide of the inaccuracy of the data?

-        What do the authors mean by reasonable data in the sentence “we select more reasonable data as the characteristic data (p. 4, Section 4.1, line 3)”

-        Authors should give examples for encountered problems and how they solved them.

-        Authors should justify their choices for the specified methods in this preprocessing phase.

  1. In section 4.2, the authors describe the Dual-link BiGRU network as composed with two sub-networks. They give description of the first one, dual-link feature network, but they did not describe the fully connected network representing the second part.
  2. In Figure 2, only the 6 best results are reported. What about the remaining 75 countries?
  3. There is no performance comparison between the proposed model Dual-link BiGRU and other competitors.

Miscellaneous

1- In Section 4, correct “constructing Dual-link BiGRU to predict COVID-19”.

2- In section 4.2, possibly replace “This paper constructs Dual-link BiGRU networks” by “In this paper, we construct ..”.

3- The authors frequently use the wording “this paper selects”, “this paper uses”. They might find it useful to replace these by wording like “In this paper, we select”, “In this paper, we use”

4- In Table 3, there are 49 effective models for Dual-link BiGRU instead of 48.

5- In section 5.2, the authors write “for the 61 selected effective national model”. Is it 61 or 81?

Conclusion

I think that the findings of the paper are a confirmation of known undertaken measures by the country to prevent the spread of the virus: the reported results are not novel. The contribution of the paper lies in the fact that the constructed model gave results supported by observed evidence.
